# Common Mitochondrial Targets of Curcumin and Cinnamic Acid, the Membrane-Active Natural Phenolic Compounds

**DOI:** 10.3390/pharmaceutics16101272

**Published:** 2024-09-29

**Authors:** Tatiana A. Fedotcheva, Natalia V. Beloborodova, Nadezhda I. Fedotcheva

**Affiliations:** 1Science Research Laboratory of Molecular Pharmacology, Medical Biological Faculty, Pirogov Russian National Research Medical University, Ministry of Health of the Russian Federation, Ostrovityanova St. 1, Moscow 117997, Russia; tfedotcheva@mail.ru; 2Federal Research and Clinical Center of Intensive Care Medicine and Rehabilitology, Petrovka St., 25-2, Moscow 107031, Russia; nvbeloborodova@yandex.ru; 3Institute of Theoretical and Experimental Biophysics, Russian Academy of Sciences, Institutskaya Str., 3, Pushchino, Moscow 142290, Russia

**Keywords:** curcumin, cinnamic acid, mitochondria, swelling, mitochondrial permeability transition pore, uncoupling, dehydrogenases, membrane potential, microbiota

## Abstract

**Background:** Research has shown the multiple actions of curcumin on different cell systems, including enzymes and mitochondria. The detected effects of curcumin on mitochondria are diverse, ranging from protective to toxic. **Objectives:** In this present work, the influence of curcumin, as well as cinnamic acid, which is a microbial metabolite and a possible product of the microbial breakdown of curcumin, on isolated mitochondria, was investigated. **Methods:** Membrane potential, swelling, respiration, and calcium retention capacity were studied using selective electrodes, fluorescence and spectral methods. **Results:** It was found that curcumin at low concentrations (10–20 μM) activated the opening of the calcium-dependent permeability transition pore (mPTP) and decreased the calcium retention capacity and threshold concentrations necessary for the mPTP opening. Moreover, curcumin caused a concentration-dependent stepwise decrease in the membrane potential, accompanied by the activation of respiration and a decrease in oxidative phosphorylation, which indicates that curcumin is a typical mitochondrial uncoupler. The uncoupling effect strongly depended on the concentration of curcumin, which also increased, stepwise, from weak uncoupling at 25 µM to complete uncoupling at 75–100 µM. Cinnamic acid had similar effects, with the exception of the depolarizing effect, at concentrations that were an order of magnitude higher. **Conclusions:** Presumably, the uncoupling action of curcumin is a priming event that modulates any energy- and redox-dependent mitochondrial functions, from positive stimulation to toxic disorder. This effect can also underlie the curcumin-induced changes in different cellular processes and be achieved by targeted delivery of curcumin to certain cells, bypassing the microbiota.

## 1. Introduction

In the search for new ways and possibilities of influencing mitochondria, natural compounds such as curcumin and cinnamic acid deserve special attention. These compounds have similar aromatic structures and are active phenolic compounds. Their positive effects are noted in pathological conditions and diseases of various organs, including the heart, lungs, kidneys, and brain [1,2,3,4].

Various effects of curcumin on physiological systems have been found, including effects on the signaling pathways, inflammatory processes, antioxidant defenses, etc. [5,6]. Recent data also indicate an unusually high activity of this natural phenolic compound [7,8]. It has been intensively studied as a natural anticancer agent that induces apoptosis and prevents multidrug resistance in cancerous cells while simultaneously demonstrating low toxicity toward normal cells [9,10,11]. One of the main features of curcumin is its ability to affect the key enzymes regulating the cellular redox balance. Curcumin was shown to affect the expression and activity of the antioxidant enzymes glutathione peroxidase, catalase, and superoxide dismutase [12,13], as well as the activity of the pro-inflammatory enzymes cyclooxygenase-2 and lipoxygenase [14,15]. By modulating their activity, curcumin can influence redox homeostasis as a whole. In particular, it inhibits the activity of glutathione peroxidase, provoking cell death through ferroptosis [16,17]. As noted, curcumin can cause opposite effects on the redox balance. It is believed that curcumin exhibits both pro-oxidant and antioxidant properties, the manifestation of which depends mainly upon its concentration and the cell type. Some studies showed that curcumin prevents ROS formation at low doses and increases it at high concentrations [7,12,17].

Another property that enables the broad impact of curcumin is its influence on mitochondrial functions. This influence is considered in the context of the induction of mitochondria-dependent apoptosis in cancer cells. It was shown that curcumin induced mitochondria-mediated ROS production and stimulated the expression of the redox-sensitive pro-apoptotic factor p53 in fibroblasts [18]. Also, curcumin caused the death of melanoma cells by affecting the mitochondrial permeability transition pore (mPTP) opening [19]. The anticancer activity of curcumin associated with mitochondrial dysfunction was demonstrated in acute lymphoblastic leukemia cells [20]. On the contrary, according to some data, curcumin has a cardioprotective effect through the prevention of mitochondrial damage and inhibition of mPTP opening [21,22]. This protective effect is usually explained by an activation of the expression of antioxidant proteins, which reduces mitochondrial oxidative stress [23,24].

Previously, we investigated the influence of the phenolic acids that are formed during the fermentation of aromatic amino acids and polyphenols by intestinal microflora (cinnamic, benzoic, phenylacetic, and phenyllactic acids, and others) on ROS formation in mitochondria and neutrophils, the main producers of ROS in tissues and circulation [25]. According to those data, cinnamic acid has a pro-oxidant effect on mitochondria and inhibits ROS formation in neutrophils. It is important to note that cinnamic acid is a precursor and a structural building block of the curcumin molecule [26]. Cinnamic acid is a product of phenylalanine ammonia-lyase, a ubiquitous higher-plant enzyme that catalyzes the nonoxidative deamination of phenylalanine to cinnamic acid. This enzyme is also found in a few bacteria, namely in *Streptomyces maritimus* and *Sorangium cellulosum*, where it is involved in anaerobic benzoyl-CoA biosynthesis, as well as in *Streptomyces verticillatus,* where it has been implicated in the biosynthesis of cinnamamide [27,28,29]. Among the microbial phenolic acids tested, cinnamic acid was the most active in influencing mitochondria. An example is the decrease in the pyridine nucleotide redox state. In this case, the phenolic acids arranged in the order of decreasing influence were (in this case, the phenolic acids were arranged in the order of decreasing effect) as follows: cinnamic acid > benzoic acid > phenylacetic acid > phenyllactic acid [30]. The goal of this work was to elucidate the effect of curcumin on the mitochondrial functions underlying mitochondrial-dependent cellular processes, such as mPTP-associated cell death, and to compare its effects with those of cinnamic acid, which is a microbial metabolite and a structural building block of the curcumin molecule and a possible product of its microbial breakdown. We studied the influence of curcumin and cinnamic acid on mitochondrial functions, including the mPTP opening, the activity of mitochondrial dehydrogenases, respiration, and oxidative phosphorylation.

## 2. Materials and Methods

### 2.1. Reagents and Chemicals

Curcumin was obtained from ChemCruz, Santa Cruz Biotechnology, Inc. (Dallas, TX, USA). All other reagents (cinnamic acid, butylhydroxytoluene, NAD, NAD(P)H, adenosine diphosphate, 2,6-dichlorophenolindophenol, phenazine metasulfate, sucrose, and others) were from the Sigma–Aldrich Corporation (St. Louis, MO, USA).

### 2.2. Preparation of Rat Liver Mitochondria

This study was conducted in accordance with the ethical principles formulated in the Helsinki Declaration on the care and use of laboratory animals. Manipulations were carried out by the certified staff of the Animal Department of the Institute of Theoretical and Experimental Biophysics (Russian Academy of Sciences and approved by the Commission on Biomedical Ethics of ITEB RAS (N1/2024, 18 March 2024). Mitochondria were isolated from adult Wistar male rats. Mitochondria from the liver were isolated using the differential centrifugation method. The liver was homogenized in an ice-cold isolation buffer containing 300 mM sucrose, 1 mM EGTA, and 10 mM HEPES–Tris (pH 7.4), and the homogenate was centrifuged at 600× *g* for 7 min at 4 °C. The supernatant fraction was centrifuged at 9000× *g* for 10 min. The pellet containing mitochondria was collected and washed twice in the same medium but without EGTA. The final mitochondrial pellet was suspended in the washing medium to yield 60–80 mg of protein/mL.

### 2.3. Assay of Swelling of Mitochondria

The swelling of mitochondria was measured at a wavelength of 540 nm using an Ocean Optics USB4000 spectrophotometer (Ocean Optic, Dunedin, FL, USA). The swelling was induced by CaCl_2_ addition and was assessed by the changes in optical density during incubation with the tested compounds. Mitochondria at a concentration of mitochondrial protein of 0.3–0.4 mg/mL were incubated in the buffer containing 125 mM KCl, 15 mM HEPES, 1.5 mM phosphate, and 5 mM substrate.

### 2.4. Determination of Mitochondrial Membrane Potential and Calcium Retention Capacity

The difference in the electric potential on the inner mitochondrial membrane was measured from the redistribution of lipophilic cation tetraphenylphosphonium (TPP^+^) between the incubation medium and mitochondria. The concentration of TPP^+^ in the incubation medium was recorded by a TPP^+^ selective electrode (Nico, Moscow, Russia). The incubation of mitochondria was carried out in an open cuvette with continuous stirring. The mitochondrial calcium retention capacity (CRC) was defined as the total concentration of added Ca^2+^ required for pore opening. The opening of the mPTP was induced by the sequential additions of CaCl_2_ to the incubation medium. The opening of the mPTP was registered as a rapid rise in the concentration of calcium ions in the incubation medium. The concentration of Ca^2+^ was registered by a Ca^2+^-selective electrode in an open cuvette of a 1 mL volume under continuous stirring. 

### 2.5. Assay of Oxidation of Succinate and NAD-Dependent Substrates by the Methyl Thiazolyl Tetrazolium (MTT) Assay and 2,6-Dichlorophenolindophenol (DCPIP)

The samples containing 125 mM KCl, 20 mM HEPES, pH 7.4, and 150 µM MTT were placed in a series of spectrophotometric cuvettes. Oxidation substrates and test compounds were added to the cuvette before the mitochondria. The acceptor reduction was initiated by the addition of mitochondria (0.5 mg of protein per mL). After 5 min of incubation, the mitochondria were lysed by Triton X-100 (10 µL of 10% solution), and the optical density was immediately recorded at 580 nm using a USB4000 spectrophotometer (Ocean Optic, Dunedin, FL, USA).

The activity of SDH was measured by the reduction of the electron acceptor DCPIP. Mitochondria (0.5 mg protein/mL) were incubated in 2 mL of a medium containing 125 mM KCl, 15 mM HEPES, and pH 7.4 in the presence of 1 mM cyanide, 10 µL of 10% triton X-100, and 200 µM DCPIP. The DCPIP reduction was induced by the addition of 5 mM succinate and then was activated with 250 µM phenazine metosulfate (PMS). The acceptor reduction rate was measured at a wavelength of 600 nm using an Ocean Optics USB4000 spectrophotometer.

### 2.6. Determination of the Redox State of Pyridine Nucleotides and Oxidative Phosphorylation in Mitochondria

The redox state of the pyridine nucleotides in mitochondria was determined by recording the fluorescence of the pyridine nucleotides (excitation at 340 nm, emission at 460 nm) on a Hitachi-F700 fluorimeter (Hitachi, Tokyo, Japan). Mitochondria (0.6 mg protein/mL) were incubated in the medium containing 125 mM KCl, 1.5 mM KH_2_PO_4,_ and 15 mM HEPES–Tris (pH 7.25), and glutamate with malate or pyruvate with malate were added as the substrates of oxidation. Adenosine diphosphate (ADP) (100 M) was added to evaluate the oxidative phosphorylation. Complete oxidation of the pyridine nucleotides was induced by adding the uncoupler, FCCP (2 µM).

### 2.7. Determination of Respiration Rates

The oxygen consumption in a mitochondrial suspension was determined by the polarographic method with a Clark-type electrode in a closed chamber of 2 mL containing 2.0 mg of mitochondrial protein under continuous stirring. Mitochondrial respiration was supported by succinate (5 mM) or glutamate (4 mM) plus malate (2 mM) and pyruvate (4 mM) plus malate (2 mM). Respiration was activated by ADP (200 M) for the evaluation of phosphorylating respiration. The respiratory control index was defined as the ratio of the ADP-stimulated respiration rate to the respiration rate after ADP phosphorylation.

### 2.8. Statistical Analysis

The data given represent the means standard error of means (SEM) from five to seven experiments or are the typical traces of three to five identical experiments with the use of different mitochondrial preparations. The statistical significance was estimated by the Student’s *t*-test with a *p* < 0.05 as the criterion of significance.

## 3. Results

### 3.1. Influence of Curcumin on Mitochondrial Permeability Transition Pore (mPTP) Opening

Figure 1 shows the influence of curcumin on the opening of the mitochondrial pore by calcium ions, which is determined by the rate of mitochondrial swelling and the changes in the threshold calcium concentration required to induce the mPTP opening. The main criterion for assessing the mitochondrial swelling as the opening of the mPTP is its inhibition by cyclosporine A (CsA). The effects of the mPTP inhibitors, CsA, as well as ADP, and the lipid radical scavenger BHT on the swelling rate in the presence of curcumin, were examined. As shown in Figure 1a, curcumin itself did not induce any swelling but strongly activated the rate of swelling induced by the calcium ions. Even at a low concentration of 10 µM, it increased the rate of swelling by more than three times. This effect was completely prevented by the mPTP inhibitor CsA and was largely reversed by BHT, while ADP, another inhibitor of the mPTP opening, had no influence on the swelling activated by curcumin (Figure 1b).

As shown in Figure 1c, increasing the concentration of curcumin led to a strong decline in the calcium retention capacity, which was manifested in a decrease in the threshold calcium ion concentrations that induced the mPTP opening. When the curcumin concentration was increased from 20 to 100 μM, a more than twofold decrease in the threshold calcium concentration was observed. The rate of calcium accumulation also slowed after each subsequent supplement. BHT partially restored the calcium retention capacity at a concentration of curcumin of 50 μM (Figure 1d) but had little influence at a curcumin concentration of 100 μM (Figure 1e). In the latter case, CsA was also ineffective (Figure 1e). As seen, calcium is not accumulated by mitochondria after two additions, and the mPTP opening does not occur. The contribution of depolarization and its role in the mPTP opening were examined in the next experiments.

### 3.2. Influence of Curcumin on the Membrane Potential and Activity of Dehydrogenases

The influence of curcumin on the membrane potential in mitochondria was evaluated by examining the changes in the TTP+ concentration in the incubation medium using a TTP+-selective electrode. As shown in Figure 2a, successive additions of curcumin caused a decrease in the membrane potential maintained by succinate oxidation. Curcumin induced a stepwise depolarization, which showed up as a decrease in as well as a stabilization of the membrane potential after each supplementation. The influence of curcumin on the membrane potential was not suppressed by CsA and slightly decreased in the presence of BHT. A similar effect of curcumin on the membrane potential was observed during the oxidation of NAD-dependent substrates. In this case, CsA did not prevent depolarization, and BHT slightly maintained the membrane potential during the curcumin supplementation (Figure 2b). The protective effect of BHT varied in the range of 20–30%, which indicates the corresponding contribution of lipid peroxidation to this process (Figure 2c). In addition, depolarization was observed upon the oxidation of both succinate and glutamate with malate. Hence, this effect of curcumin is independent of the type of oxidation substrate.

In the following experiments, we examined the influence of curcumin on the activity of succinate dehydrogenase (SDH) and NAD-dependent dehydrogenases. The activity of SDH was estimated spectrophotometrically from the reduction in the electron acceptor dichlorophenolindophenol (DCPIP). As shown in Figure 2d, curcumin in the concentration range of 10–100 μM had no effect on SDH activity. The inhibition of SDH by malonate, a selective SDH inhibitor, served as a control of the specificity of the measurement.

The activity of NAD-dependent dehydrogenases was determined from the fluorescence of reduced NADH during the oxidation of glutamate and malate by frozen/thawed mitochondria. At a concentration of 20 μM, curcumin decreased the rate of NAD reduction during glutamate oxidation by three times (Figure 2e). However, the control measurements showed that curcumin by itself suppressed NADH fluorescence in a concentration-dependent manner (Figure 2e, insert), making the adequate interpretation of these data difficult. To avoid this effect, in the following experiments, we used the colorimetric method with MTT as an electron acceptor. In each series of experiments, the contribution of the oxidation of endogenous substrates to the MTT reduction was assessed. This value includes all side oxidative reactions and is negligible compared to the MTT reduction by an added substrate, glutamate, or succinate (Figure 2e). As shown in Figure 2e, curcumin at concentrations of 10–20 µM decreased the MTT reduction supported by glutamate oxidation when NADH oxidation through the respiratory chain was inhibited by rotenone. Under these conditions, the actual activity of glutamate dehydrogenase was evaluated. Curcumin decreased GDH activity by 15% and almost by 40% at concentrations of 5 and 10 μM, respectively. Curcumin also diminished the oxidation of pyruvate, another NAD-dependent substrate. In this case, at a concentration of 10 μM, curcumin caused a 25% inhibition of MTT reduction. As in the experiment with DCPIP, curcumin did not affect the reduction of MTT during succinate oxidation (Figure 2e). The selective SDH inhibitor malonate was also used as a relevant control.

Since curcumin does not change the SDH activity, the depolarization is not related to the enzyme inhibition and, therefore, is due to the unique protonophore activity of curcumin. Figure 2f shows that, indeed, at high curcumin concentrations, calcium did not induce swelling, although the ability to swell was retained, as evidenced by its activation upon the addition of alamethicin—a peptide that forms the nonspecific channel of permeability of mitochondrial membranes. The lack of the effect of calcium on swelling can be explained by depolarization, which prevents the voltage-dependent entry of calcium ions, as shown in Figure 2f, inset.

These assumptions were verified in the following experiments, in which the influence of curcumin on respiration and oxidative phosphorylation was estimated by the polarographic method.

### 3.3. Influence of Curcumin on Mitochondrial Respiration and Oxidative Phosphorylation

To evaluate the uncoupling effect on mitochondria, we examined the influence of curcumin at different concentrations on respiration and oxidative phosphorylation. As shown in Figure 3a, curcumin strongly activated respiration during succinate oxidation. The activation of respiration was highly dependent on the concentration of curcumin. With subsequent additions of curcumin, 25 μM each, the rate of respiration increased stepwise, from 50% activation after the first addition to three- to fourfold activation when the concentration was increased to 50–75 μM. A further increase in the concentration either did not stimulate or only slightly activated the rate of respiration.

As shown in Figure 3b, the activation of respiration was accompanied by a decrease in the efficiency of oxidative phosphorylation, as assessed by the respiratory control index, which was calculated as the ratio of respiration rates in the course of and after ADP phosphorylation. At a low concentration of 25 µM curcumin, a moderate decrease, by about 20–30%, in respiratory control was observed. At a concentration of 50 µM, the respiratory control decreased by three times and almost completely disappeared at a concentration of 75 µM. The artificial uncoupler FCCP was added to achieve zero oxygen in the sample. Being one of the most powerful uncouplers, it caused an additional stimulation of respiration. Thus, the uncoupling effect of curcumin on respiration supported by succinate oxidation is obvious, and its magnitude is determined only by the concentration of curcumin (Figure 3c,d).

A similar stepwise increase in the respiration rate with an increasing curcumin concentration also occurred during the oxidation of the NAD-dependent substrates glutamate with malate. However, in this case, exceeding the concentration of curcumin above 50 μM led to a decline in respiration, which could be reactivated by the addition of succinate (Figure 4a). A decrease in the respiration rate was observed both with sequential additions of curcumin and its immediate addition at a high concentration (Figure 4a, insert). The removal of inhibition by succinate indicates that this inhibition is specific for NAD-dependent substrates and is not due to the inhibition of the mitochondrial respiratory chain. These data are also consistent with those obtained by the MTT assay. Simultaneously with the activation of respiration, a decrease in oxidative phosphorylation was observed (Figure 4b).

Curcumin caused similar changes during the oxidation of pyruvate with malate; namely, it activated respiration and declined oxidative phosphorylation at concentrations up to 50 μM, and induced the inhibition of respiration at higher concentrations (Figure 4c,d). Figure 4e,f show the dependence of the respiration rate and respiratory control on the curcumin concentration in the range of 25–100 μM.

Thus, the uncoupling action of curcumin strongly depends on its concentration and is manifested in the activation of respiration and a decrease in oxidative phosphorylation. Moreover, at high concentrations, curcumin decreases NAD-dependent respiration.

### 3.4. Influence of Cinnamic Acid on Mitochondrial Functions

As we observed earlier, cinnamic acid exhibited curcumin-like properties [25,30]. We compared the action of cinnamic acid and curcumin on the MPTP opening, the membrane potential, and respiration. Figure 5 shows the influence of cinnamic acid on the calcium-induced mPTP opening. At concentrations of 100 and 200 μM, cinnamic acid activated the rate of calcium-induced swelling by 1.5 and almost 2 times, respectively (Figure 5a). This effect was eliminated in the presence of BHT, a lipid radical scavenger. Moreover, cinnamic acid lowered resistance to the calcium ion loading and pore opening. This effect was strongly dependent on the oxidation substrate, whereas, during the oxidation of succinate, cinnamic acid only caused a 20% decrease in the calcium retention capacity (Figure 5b). The decrease during the oxidation of NAD-dependent substrates was more than 50% (Figure 5c). BHT had a protective effect on the calcium load, mainly during the oxidation of succinate, and did not show any pronounced influence under the oxidation of glutamate with malate or pyruvate with malate. The greatest decrease in the threshold calcium concentrations inducing pore opening was observed during the oxidation of pyruvate with malate (Figure 5d).

Also, cinnamic acid provoked a slow decline in the membrane potential during the oxidation of pyruvate with malate without affecting it during the oxidation of other substrates (Figure 6a). The extent to which this effect may be related to the oxidation of NADH or the activity of NAD-dependent dehydrogenases was examined by NADH fluorescence and MTT reduction assays. As shown in Figure 6b,c, cinnamic acid induced the oxidation of NADH and decreased the efficiency of oxidative phosphorylation. These alterations were observed during the oxidation of both NAD-dependent substrates, glutamate and pyruvate, with more pronounced changes upon the oxidation of pyruvate. In both cases, the addition of cinnamic acid slowed down the rate of NAD reduction and led to incomplete NAD reduction after ADP phosphorylation compared to the corresponding controls for these substrates. According to the data obtained by the fluorescence assay, cinnamic acid decreased the phosphorylation efficiency by 20% and 50% upon glutamate and pyruvate oxidation, respectively. The induction of NADH oxidation by cinnamic acid was more pronounced during the oxidation of pyruvate than glutamate and was only weakly inhibited by BHT (Figure 6d). The MTT assay showed that the contribution of NAD-dependent dehydrogenases to these changes is either minimal or absent. In the presence of rotenone, a respiratory chain inhibitor, cinnamic acid had no effect on glutamate oxidation and weakly decreased pyruvate oxidation (Figure 6e). Also, cinnamic acid did not affect the activity of succinate dehydrogenase, as shown by the assay with DCPIP as an acceptor (Figure 6f).

Respiration measurements showed that cinnamic acid (200 µM) did not affect the respiration and oxidative phosphorylation during succinate oxidation (Figure 7a) and weakly diminished the rate of FCCP-activated respiration during the oxidation of NAD-dependent substrates (Figure 7b). The respiration decline was 20–30% at a cinnamic acid concentration of 200 μM.

Table 1 summarizes the data on the effects of curcumin and cinnamic acid on mitochondrial functions and possible targets. The common effects of the compounds on mitochondria are the stimulation of the opening of MPTP by calcium ions, the activation of lipid peroxidation, and the partial inhibition of the activity of NAD-dependent dehydrogenases. The fundamental differences between them are that curcumin decreases the membrane potential, which is related to its protonophore activity—a feature not characteristic of cinnamic acid—as well as their effective concentrations, which for cinnamic acid is one order of magnitude higher than that for curcumin.

## 4. Discussion

Curcumin has diverse and numerous effects on various physiological systems, which are often associated with their direct or indirect influence on mitochondrial functions. The current evidence indicates that mitochondria are the key targets of curcumin and that the changes in mitochondrial functions may determine subsequent events in cells and organs. It was shown that curcumin has both protective and toxic effects on mitochondrial functions [19,20,21,22]. Our study revealed that curcumin affects interdependent membrane processes, including concentration-dependent stepwise depolarization, respiration activation, and stimulation of the mPTP opening by calcium ions. All three processes are related to changes in the membrane processes, which allows one to consider curcumin as a membrane-active compound. As shown by our data, the depolarization is accompanied by a strong activation of respiration and a decrease in oxidative phosphorylation. Similar to depolarization, the uncoupling effect of curcumin strongly depends on its concentration, increasing also stepwise from weak uncoupling at 25 µM to complete at 75–100 µM. It can be assumed that the uncoupling action of curcumin is a priming event that affects any energy- and potential-dependent changes in mitochondrial functions. Cinnamic acid exhibited a similar action on the mitochondrial processes, except for depolarization and uncoupling. A comparison of curcumin with cinnamic acid is of interest, considering that cinnamic acid is both a precursor in the biosynthesis of curcumin and its possible product during microbial degradation. Another important aspect is the application of both compounds in the synthesis of new pharmaceuticals [11,31,32,33]. Moreover, cinnamic acid is considered an attractive building block for the development of pharmacological tools [34].

The fact that the effects of curcumin are highly concentration-dependent is in agreement with the data showing that the effect of curcumin varies from protective to toxic as its concentration increases. According to the available data, this dependence is especially pronounced when examining the effects of curcumin on oxidative stress, the mPTP opening, and mitochondria-dependent apoptosis. Thus, it was shown that curcumin has both pro-oxidant and antioxidant properties, preventing ROS formation at low doses (5 µM) and increasing it at high (25 µM) concentrations [12]. Also, curcumin at a high concentration (70 µM) induced an elevation in ROS production and cell death incidence in acute lymphoblastic leukemia [20]. It was also found that curcumin activated mitochondria-mediated ROS formation and induced the apoptosis of fibroblasts [18]. The protective effect of curcumin is usually explained by an enhancement of the expression of antioxidant enzymes that reduce oxidative stress in injured mitochondria. Moreover, it was attributed to the improvement of mitochondrial function in neurological, cardiovascular, and some other diseases [13,15,23,24].

As is known, curcumin also had dual effects on the mPTP opening and apoptosis. Thus, it induced mitochondrial pore opening and apoptosis in melanoma cells [19] and prevented these processes in heart mitochondria [21]. It was assumed that curcumin promotes the mPTP opening through the reduction of Fe^3+^ to Fe^2+^, hydroxyl radical production, and the oxidation of thiol groups in the membrane [35,36].

It is believed that the dominant property that determines the diverse effects of curcumin is its protonophoric activity [37]. Our data, obtained on standard isolated rat liver mitochondria, are in good agreement with this concept, demonstrating that the degree of depolarization, depending on the concentration of curcumin, directly regulates the calcium retention capacity of mitochondria and the threshold concentrations of calcium ions required for the opening of mPTP. At low concentrations, curcumin causes a partial decrease in the membrane potential, which leads to a decrease in voltage-dependent calcium accumulation and the mPTP opening at calcium concentrations lower than in the control. At high concentrations of curcumin, the strong depolarization blocks the entry of calcium ions into mitochondria, which manifests itself in that calcium ions do not induce swelling. In this case, the calcium-induced mPTP does not open, although the ability of mitochondria to swell remains since swelling is activated by alamethicin, which forms a nonspecific channel of permeability in mitochondrial membranes. These data may explain the known opposite effects of curcumin on the mPTP opening as a mechanism of cell death.

It is important to emphasize that depolarization decreases the level of oxidative stress in mitochondria, which may also explain the conflicting data on the effect of curcumin on oxidative stress. According to some data, ROS production is highly sensitive to the membrane potential. It was shown that even a small decrease in the membrane potential, on the order of 10 mV, was sufficient to abolish around 70% of the ROS produced by mitochondria [38]. It was also noted that mild mitochondrial uncoupling protects cells from death by attenuating oxidative stress and mitochondrial damage [39,40]. In this context, the uncoupling effect may be one of the reasons for the antioxidant activity of curcumin. Nevertheless, our data show that the lipid radical scavenger, BHT, lowered the rate and the amplitude of swelling activated by curcumin. Also, BHT partially returned the threshold calcium concentrations required for the opening of mPTP and increased the calcium retention capacity. It can be assumed that lipid peroxidation is either a side effect of curcumin as a membrane-active compound or is a consequence of total oxidation, including that of reducing agents, as a result of uncoupling. This effect may be involved in the implementation of the pro-oxidant activity of curcumin and, in particular, the activation of ferroptosis associated with lipid peroxidation in cytoplasmic and mitochondrial membranes [41]. As was recently found, curcumin also suppressed the expression of glutathione peroxidase, which led to excessive lipid peroxidation in cancer cells compared to the control [42].

Our data show that curcumin does not affect the activity of SDH and diminishes the activity of NAD-dependent dehydrogenases. This made itself especially evident as a decrease in the respiration rate during the oxidation of NAD-dependent substrates with increasing concentrations of curcumin above 50 μM. The effect of curcumin on the activity of NAD-dependent respiration may be a particular factor in the regulation of mitochondrial functions. This suggests that the influence of curcumin may expand to the energization of mitochondria supported by a particular substrate. Some data on this topic were obtained on endometrial cells, where curcumin inhibited complex I of the respiratory chain and decreased the activity of oxidative phosphorylation [43]. In addition, curcumin was shown to induce a burst of SDH activity, excessive ROS production, and mitophagy in thyroid carcinoma cells [44]. It can be assumed that the diverse influence of curcumin on cell mitochondria is due to their different intrinsic functional state, while the influence on isolated control mitochondria is determined only by the physico-chemical properties of the compound. It is assumed that some redox sites of curcumin (hydroxy and methoxy groups, keto-enol tautomer) are involved in various redox reactions in mitochondria [15,24,37].

Summarizing the above data and our results, it can be assumed that curcumin at low concentrations, up to 50 µM, has a positive, stimulating effect by activating mitochondrial functions, especially respiration, due to a moderate uncoupling effect. In contrast, at higher concentrations above 50 µM, negative consequences of deeper oxidation, including that of thiol- and NAD-dependent reductants, may be induced.

Cinnamic acid (or phenylacrylic acid) is also a redox-active compound and exhibits properties similar to those of curcumin with the exception of protonophore activity. Previously, we have shown that cinnamic acid activates ROS production in mitochondria and inhibits it in neutrophils, and these effects are mediated via the interaction with thiol groups [25,45]. Consequently, cinnamic acid can participate in the regulation of ROS production in both the circulation and tissues, changing the level of oxidative stress. Cinnamic acid is the most active among other phenolic acids formed during the fermentation of aromatic amino acids and polyphenols by microbiota.

As follows from our data, cinnamic acid activates the mPTP opening by calcium ions, stimulates lipid peroxidation, and decreases the activity of NAD-dependent dehydrogenases. Its action on membrane permeability was almost completely eliminated by BHT, an inhibitor of lipid peroxidation. BHT is a known strong radical-trapping antioxidant, which blocks lipid autoxidation by converting the peroxyl radicals to non-radical products [46]. Moreover, the effects of cinnamic acid depended, to a great extent, on the energy state of mitochondria. This dependence was especially pronounced in the resistance to the calcium ion loading and pore opening—high during the oxidation of succinate and low during the oxidation of NAD-dependent substrates—observed in the presence of cinnamic acid. These observations are consistent with the data on the inhibitory effect of cinnamic acid on NAD-dependent respiration and the lack of this effect on respiration supported by succinate oxidation.

In addition to the fact that cinnamic acid is simultaneously a building block of curcumin and a microbial metabolite, an interesting relationship between them can be traced in the context of the role of the microbiota in their biotransformation. A recent study showed that curcumin was able to exert beneficial effects via the “brain-gut” axis [47]. Moreover, it markedly changed the overall composition of the gut microbiota, modulating more than one hundred operational taxonomic units [48]. Microbial transformation was recently applied to obtain new derivatives of cinnamic acid with neuroprotective properties [49]. It is important to note that curcumin and cinnamic acid have structural similarities with aromatic microbial metabolites of the human microbiota of a healthy person, the deficiency of which is associated with multiple organ dysfunctions [50,51]. The similarity of the mitochondrial dysfunctions shows promise for the testing and probably application of curcumin and cinnamic acid in various pathologies. Their biotransformation by the microbiota suggests that both compounds can serve as effective prebiotics.

It should be noted that the use of selective electrodes, in contrast to fluorescence and spectral methods, makes it possible to test curcumin in a wide range of concentrations. The resulting effects can underlie the curcumin-induced changes in different cellular processes and be achieved by a targeted delivery of curcumin to certain cells, bypassing the microbiota.

## 5. Conclusions

Curcumin and cinnamic acid are membrane-active compounds that sensitize mitochondria to open mPTP. The uncoupling action of curcumin is a priming event that determines subsequent changes dependent on the membrane potential and the redox state, including a decrease in resistance to the calcium load and in the threshold calcium concentrations inducing the mPTP opening. Presumably, various effects of curcumin are determined by its moderate protonophore activity, while the influence of cinnamic acid is mainly associated with the stimulation of lipid peroxidation in membranes. Our data support the assumptions that at low concentrations, curcumin can provide a positive, stimulating influence, but at high concentrations, it can provoke toxic effects. The data indicate the role of curcumin in modifying the mitochondrial functions associated with cell survival or death. However, the implementation of these effects of curcumin in cells requires further clarification.

## Figures and Tables

**Figure 1 pharmaceutics-16-01272-f001:**
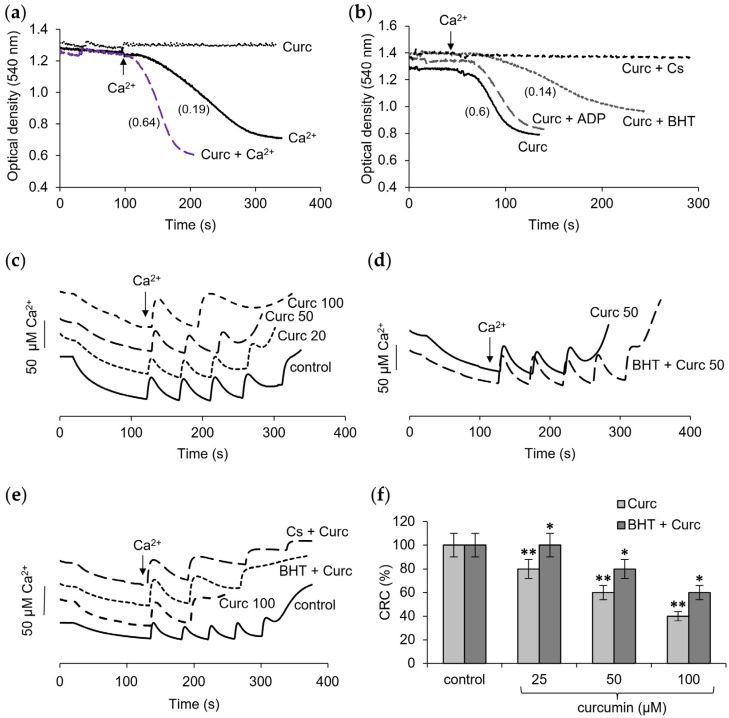
Influence of curcumin on mitochondrial permeability transition pore (mPTP) opening. Influence of curcumin (Curc, 10 µM) on the swelling induced by calcium ions (**a**); influence of mPTP inhibitors CsA (1 µM), ADP (25 µM), and lipid peroxidation inhibitor BHT (10 µM) on swelling activated by curcumin (**b**); effect of curcumin at different concentrations on calcium accumulation (**c**) and influence CsA (1 µM) and BHT (10 µM) on the calcium accumulation in the presence of 50 µM curcumin (**d**) and 100 µM curcumin (**e**); the decrease of calcium retention capacity (CRC) at different concentrations of curcumin and influence of BHT (**f**). Swelling rates are indicated in parentheses. All measurements were carried out in a buffer containing 125 mM KCl, 15 mM HEPES, 1.5 mM phosphate, and 5 mM succinate at a protein concentration of 0.35 mg/mL in the optical density assay and 1.5 mg/mL in calcium selective electrode assay; each calcium addition is equal to 50 µM. An asterisk indicates values that differ significantly from the control values (*p* < 0.05), (*)—for the Curc + BHT group, and (**)—for the Curc group.

**Figure 2 pharmaceutics-16-01272-f002:**
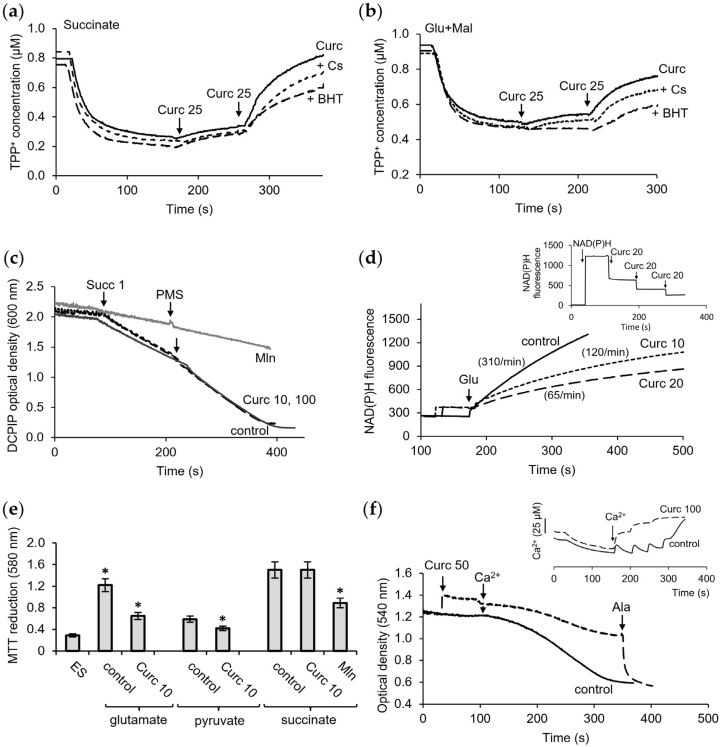
Influence of curcumin on the membrane potential and activity of dehydrogenases. Effect of sequential additions of curcumin in the indicated concentrations on the membrane potential supported by succinate (**a**) and glutamate with malate (**b**) oxidation and influence of CsA and BHT on the depolarization induced by curcumin; influence of curcumin on the activity of succinate dehydrogenase measured by DCPIP reduction assay (**c**) and glutamate dehydrogenase measured by NAD reduction, where reaction rates are indicated in parentheses and inset shows quenching NADH fluorescence by curcumin (**d**); influence of curcumin on the activity of dehydrogenases measured by MTT reduction assay (**e**); effect of depolarization on the swelling (**f**) and calcium accumulation (insert). Additions: 1 mM malonate (Mln), 5 µg alamethicin (Ala), and 50 µM CaCl_2_. ES—endogenous substrates. An asterisk (*) indicates values that differ significantly from the control values (*p* < 0.05).

**Figure 3 pharmaceutics-16-01272-f003:**
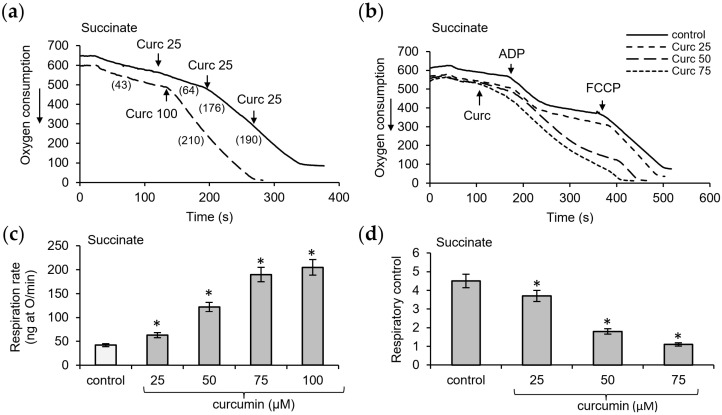
Influence of curcumin on the respiration and oxidative phosphorylation supported by succinate oxidation. Effect of curcumin in the indicated concentrations (µM) on the respiration rate (**a**) and oxidative phosphorylation (**b**); dependence of the respiration rate (**c**) and respiratory control (**d**) on curcumin concentration. Respiration rates are indicated in parentheses; all measurements were carried out in a buffer containing 125 mM KCl, 15 mM HEPES, 1.5 mM phosphate, and 5 mM succinate at a protein concentration of 2 mg/mL; 200 µM ADP and 2 µM FCCP has been added where indicated. An asterisk (*) indicates values that differ significantly from the control values (*p* < 0.05).

**Figure 4 pharmaceutics-16-01272-f004:**
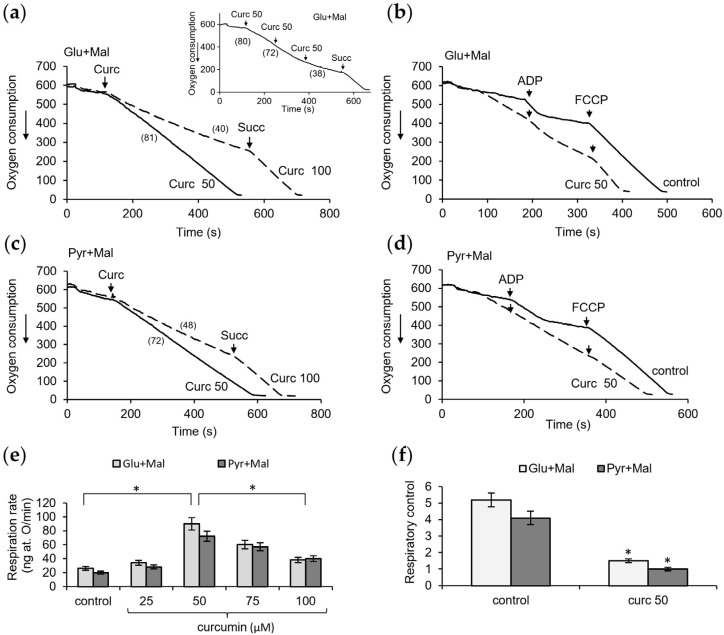
Influence of curcumin on the respiration and oxidative phosphorylation supported by NAD-dependent substrates. Effect of curcumin in the indicated concentrations (µM) on the respiration rate (**a**) and oxidative phosphorylation (**b**) during oxidation of glutamate with malate and on respiration rate (**c**) and oxidative phosphorylation (**d**) pyruvate with malate; dependence of the respiration rate (**e**) and respiratory control (**f**) on curcumin concentration. Respiration rates are indicated in parentheses; all measurements were carried out in a buffer containing 125 mM KCl, 15 mM HEPES, 1.5 mM phosphate, 4 mM glutamate or pyruvate, and 2 mM malate at a protein concentration of 2 mg/mL; 200 µM ADP and 2 µM FCCP has been added where indicated. An asterisk (*) indicates values that differ significantly from the control values (*p* < 0.05).

**Figure 5 pharmaceutics-16-01272-f005:**
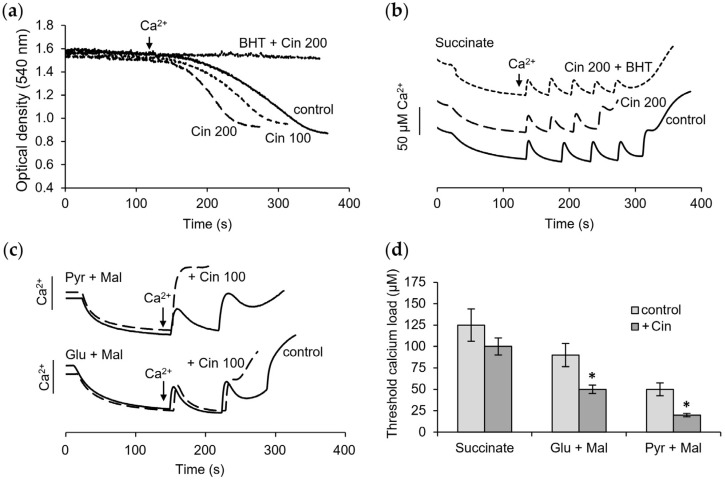
Influence of cinnamic acid on mitochondrial permeability transition pore (mPTP) opening. Influence of cinnamic acid (Cin) in indicated concentrations and BHT (10 µM) on the swelling induced by calcium ions (**a**) and the calcium accumulation (**b**) during oxidation of succinate; effect of cinnamic acid on calcium accumulation during oxidation of NAD-dependent substrates (**c**); substrate-dependent changes in threshold calcium concentrations inducing pore opening (**d**). An asterisk (*) indicates values that differ significantly from the control values (*p* < 0.05).

**Figure 6 pharmaceutics-16-01272-f006:**
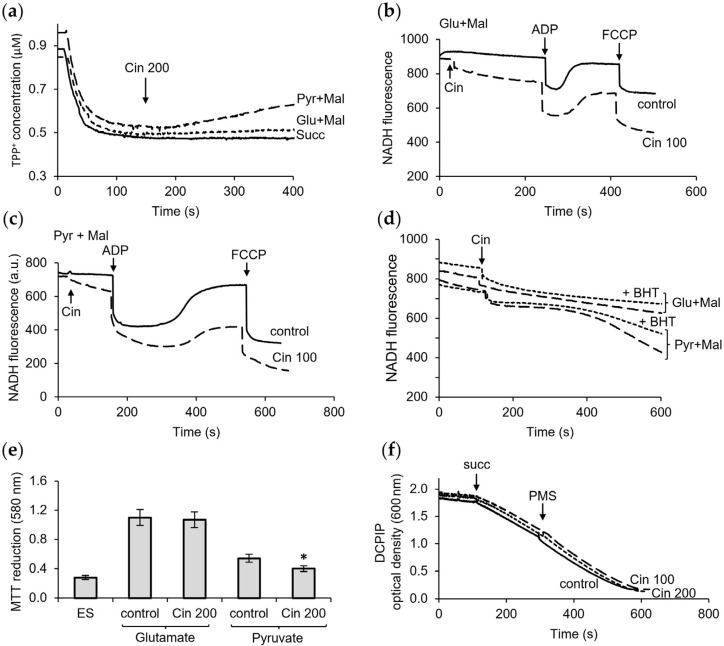
Influence of cinnamic acid on the membrane potential and activity of dehydrogenases. Influence of cinnamic acid (Cin) on the membrane potential supported by different substrates (**a**); effect of cinnamic acid on oxidative phosphorylation during oxidation of glutamate with malate (**b**) and pyruvate with malate (**c**); activation of NADH oxidation by cinnamic acid (**d**); effect of cinnamic acid on activity of NAD-dependent dehydrogenases (**e**) and succinate dehydrogenase (**f**). Additions: 200 µM ADP, 1 µM FCCP, 10 µM BHT, 1 µM rotenone (Rot), and 5 mM substrate. ES—endogenous substrates. An asterisk (*) indicates values that differ significantly from the control values (*p* < 0.05).

**Figure 7 pharmaceutics-16-01272-f007:**
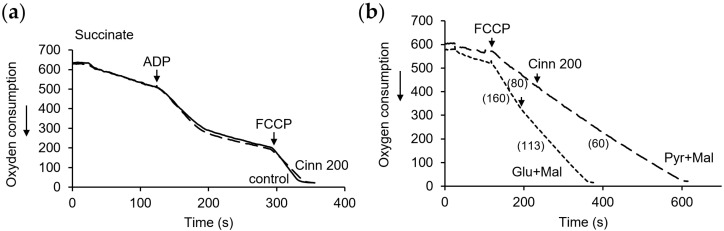
Influence of cinnamic acid on the mitochondrial respiration. Cinnamic acid (200 µM) does not affect respiration and oxidative phosphorylation during succinate oxidation (**a**) and diminishes the rate of FCCP-activated respiration during oxidation of NAD-dependent substrates (**b**). Respiration rates are indicated in parentheses. All measurements were carried out in a buffer containing 125 mM KCl, 15 mM HEPES, 1.5 mM phosphate, 5 mM succinate, or 4 mM glutamate (or pyruvate) with 2 mM malate at a protein concentration of 2 mg/mL; 200 µM ADP and 2 µM FCCP were added where indicated.

**Table 1 pharmaceutics-16-01272-t001:** The influence of curcumin and cinnamic acid on mitochondrial functions and their possible targets.

Mitochondrial Function, Target	Curcumin (10–50 µM) 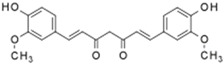	Cinnamic Acid (100–200 µM) 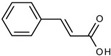
Membrane potential	depolarization	no effect
Respiration	uncoupling effect	no effect
MPTP opening by Ca^2+^	stimulation	stimulation
Resistance to calcium ion loading	decrease	decrease
Swelling rate	activation	activation
Lipid peroxidation	low activation	activation
SDH activity	no effect	no effect
GDH activity	decrease at high concentrations	decrease
PDH activity	decrease at high concentrations	decrease

## Data Availability

The datasets generated during and/or analyzed during the current study are available from the corresponding author upon reasonable request.

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
