# Peer review of "Common Mitochondrial Targets of Curcumin and Cinnamic Acid, the Membrane-Active Natural Phenolic Compounds"

_pharmaceutics, 2024, doi:10.3390/pharmaceutics16101272_

Round 1

Reviewer 1 Report (Previous Reviewer 2)

Comments and Suggestions for Authors

The manuscript is appropriately revised and it can be published in the current state.

Author Response

  1. Comments and Suggestions for Authors

The manuscript is appropriately revised and it can be published in the current state.

Thank you very much for reviewing our article.

Reviewer 2 Report (New Reviewer)

Comments and Suggestions for Authors

I suggest minor revisions for the work conducted by Fedotcheva and co-workers. It brings relevant results in an emergent topic for the international readers of Pharmaceutics.

The abstract has to be rewritten. In its current form, it is too long. According to the journal’s guidelines, the limit is 200 words. Conclusions and future directions should be clearly stated at the end of this section.

The Introduction should be expanded and the study’s need and novelty have to be better explained.

Line 106: Please, specify what the “other reagents” are.

I suggest you to provide the study strengths and limitations at the end of Discussion.

What are the practical implications of your obtained results and what should be done in future investigations? Please, mention it in your Conclusions.

Author Response

Comments and Suggestions for Authors

Thank you very much for reviewing our article. All our additions to the text are highlighted in blue.

I suggest minor revisions for the work conducted by Fedotcheva and co-workers. It brings relevant results in an emergent topic for the international readers of Pharmaceutics.

The abstract has to be rewritten. In its current form, it is too long. According to the journal’s guidelines, the limit is 200 words. Conclusions and future directions should be clearly stated at the end of this section.

We shortened the abstract to 225 words and highlighted the role of the uncoupling effect of curcumin in the modulation of mitochondrial and cellular processes, as well as the importance of the targeted delivery to certain cells, bypassing the microbiota.

Abstract: Current evidence has shown the multiple action of curcumin on different cell systems, including enzymes and mitochondria. The detected effects of curcumin on mitochondria are diverse, ranging from protective to toxic. In the present work, the influence of curcumin as well as cinnamic acid, which is a microbial metabolite and a possible product of the microbial breakdown of curcumin, on isolated mitochondria was investigated. It was found that curcumin at low concentrations (10–20 μM) activated the opening of the calcium-dependent permeability transition pore (mPTP) and decreased the calcium retention capacity and threshold concentrations necessary for mPTP opening. Besides, curcumin caused a concentration-dependent stepwise decrease in the membrane potential,  accompanied by the activation of respiration and a decrease in oxidative phosphorylation, which permits one to assign curcumin to typical mitochondrial uncouplers. The uncoupling effect strongly depended on the concentration of curcumin, increasing also stepwise from weak uncoupling at 25 µM to complete at 75–100 µM. Cinnamic acid had similar effects, with the exception of the depolarizing effect, at concentrations an order of magnitude higher. Presumably, the uncoupling action of curcumin is a priming event that modulates any energy- and redox-dependent mitochondrial functions, from positive stimulation to toxic disorders. This effect can also underlie the curcumin-induced changes in different cellular processes and be achieved by targeted delivery of curcumin to certain cells, bypassing the microbiota.

The Introduction should be expanded and the study’s need and novelty have to be better explained.

We have added the following text to Introduction:

The goal of this work was to elucidate the effect of curcumin on mitochondrial functions underlying mitochondrial-dependent cellular processes such as mPTP-associated cell death and to compare its effects with those of cinnamic acid, which is a microbial metabolite, a structural block of the curcumin molecule and a possible product of its microbial breakdown.

Line 106: Please, specify what the “other reagents” are.

We have listed the reagents:

All other reagents (cinnamic acid, butylhydroxytoluene, NAD, NAD(P)H, adenosine diphosphate, 2,6-dichlorophenolindophenol, phenazine metasulfate, sucrose, and others)

I suggest you to provide the study strengths and limitations at the end of Discussion.

We have added the following text to the end of the Discussion:

It should be noted that the use of selective electrodes, in contrast to fluorescence and spectral methods, makes it possible to test curcumin in a wide range of concentrations. The resulting effects can underlie curcumin-induced changes in different cellular processes and be achieved by targeted delivery of curcumin to certain cells, bypassing the microbiota.

 What are the practical implications of your obtained results and what should be done in future investigations? Please, mention it in your Conclusions.

We have added the following text to the Conclusions:

The data indicate the role of curcumin in modifying the mitochondrial functions associated with cell survival or death. The implementation of these mitochondrial effects of curcumin in cells requires further clarification.

This manuscript is a resubmission of an earlier submission. The following is a list of the peer review reports and author responses from that submission.

Round 1

Reviewer 1 Report

Comments and Suggestions for Authors

SUMMARY:

Authors describe the effects of curcumin and cinnamic acid (a putative precursor and/or product of curcumin metabolism) on mitochondrial swelling, oxidation of substrates, and ROS production. Manuscript preparation is largely incomplete, making evaluation of the findings presented difficult and the manuscript unacceptable for publication in its current form

MAJOR COMMENTS:The elimination of other reasons for mitochondrial swelling would need to be established before interpreting the mitochondrial swelling assay as MPTP opening. The interpretation that (line 172) “These data indicate the involvement of lipid peroxidation in the activation of mPTP opening by curcumin” needs to be explained.

Related, authors state in the introduction that they “assessed the possible involvement of lipid peroxidation and changes in the activity of dehydrogenase” in the “effects” of curcumin and cinnamic acid, but then do not show data directly relevant to lipid peroxidation

Using MTT reduction as a proxy for substrate oxidation needs to be better supported and controlled; though isolated mitochondria should eliminate non-mitochondrial oxidation, this needs to be demonstrated, or substituted with oxygraphy.

MINOR COMMENTS:

Line 34: missing “by” between “determined its”

No citation for last sentence of 1st para. of introduction. Insufficient citation throughout the intro paragraphs

Extra spaces following citation numbers need correcting

Microbial species names should be italiziced

Methods and Figures

Line 104: “by the standard methods” has no citation

Methods descriptions use inconsistent nomenclature (x g)

Inconsistent capitalization of section headings

Figure 1: What statistics were performed on these data?

Figure 2: Significance is indicated, but Student’s t-test is inappropriate when performing multiple comparisons.

Comments on the Quality of English Language

English grammar and syntax should be carefully reviewed throughout the manuscript.

Author Response

Thank you very much for your useful questions and comments. We have made corresponding corrections and additions to the manuscript (marked in red in manuscript).

The elimination of other reasons for mitochondrial swelling would need to be established before interpreting the mitochondrial swelling assay as MPTP opening.

Replay:

In our experiments, we tested the influence of curcumin and cinnamic acid on the Ca-induced opening of the mitochondrial permeability transition pore.  Typically, MPTP opening is induced by calcium ions and inhibited by CsA. The main criterion for assessing the mitochondrial swelling as the opening of MPTP is its inhibition byCsA. This result is shown in Figure 1 B, in which the swelling induced by calcium and activated by curcumin is completely inhibited by CsA.

We have made the following addition to the Results:

The main criterion for assessing the mitochondrial swelling as the opening of mPTP is its inhibition by cyclosporine A (CsA). The effects of the mPTP inhibitors, CsA as well as ADP and the lipid radical scavenger BHT, on the swelling rate in the presence of curcumin were examined.

 The interpretation that (line 172) “These data indicate the involvement of lipid peroxidation in the activation of mPTP opening by curcumin” needs to be explained.

 Replay:

BHT is the most potent inhibitor of lipid peroxidation. The fact that swelling is partially inhibited by the lipid radical scavenger BHT indicates the involvement of lipid peroxidation. This result is consistent with the known data on the effect of curcumin on the thiol groups of the membrane, since their oxidation or binding promotes oxidative processes. Previously, we have shown that the effect of cinnamic acid is also associated with interaction with thiol groups. This aspect is considered in the Discussion with relevant references.

 We have made the following addition to the Results:

Thus, the swelling induced by calcium and activated by curcumin is completely inhibited by CsA and is partially by the lipid radical scavenger BHT, which indicates the involvement of lipid peroxidation in the activation of mPTP opening by curcumin.

Related, authors state in the introduction that they “assessed the possible involvement of lipid peroxidation and changes in the activity of dehydrogenase” in the “effects” of curcumin and cinnamic acid, but then do not show data directly relevant to lipid peroxidation

Replay:

Our results show that lipid peroxidation is  involved only in the activation of mPTP. Other effects of curcumin were not associated with lipid peroxidation since they are not or very weakly affected by BHT.

To clarify the description of these data, we have made the following additions to the Results:

Thus, the curcumin-induced decrease in the membrane potential is not associated with lipid peroxidation or changes in the activity of mitochondrial dehydrogenases. However, the observed decline in the activity of NAD-dependent dehydrogenases may impair energy-dependent processes in mitochondria such as the maintenance of the membrane potential and the calcium retention capacity.

Using MTT reduction as a proxy for substrate oxidation needs to be better supported and controlled; though isolated mitochondria should eliminate non-mitochondrial oxidation, this needs to be demonstrated, or substituted with oxygraphy.

Replay:

We agree. Indeed, when measuring the MTT reduction, we always evaluate the contribution of oxidation of endogenous substrates as the control. The reduction of MTT by endogenous substrates includes all side oxidative reactions and usually amounts to 15-25% of the MTT reduction by the added substrate. Now, we have added columns with the value of MTT reduction by endogenous substrates to the diagrams..

 We have made the following amendment to the Results:

In each series of experiments, the contribution of the oxidation of endogenous substrates to the MTT reduction was assessed. This value includes all side oxidative reactions and is negligible compared to the MTT reduction by an added substrate, glutamate or succinate (Figure 2E).

MINOR COMMENTS:

Line 34: missing “by” between “determined its”

It is corrected.

No citation for last sentence of 1st para. of introduction. Insufficient citation throughout the intro paragraphs

 We have added new references (6 ref.) in the first and second paragraph of Introduction.

Extra spaces following citation numbers need correcting

Microbial species names should be italiziced

 It is corrected.

Methods and Figures

Line 104: “by the standard methods” has no citation

Methods descriptions use inconsistent nomenclature (x g)

It is corrected

Inconsistent capitalization of section headings

 It is corrected.

Figure 1: What statistics were performed on these data?

Figure 2: Significance is indicated, but Student’s t-test is inappropriate when performing multiple comparisons.

The figures show the typical traces of three to five identical experiments with the use of different mitochondrial preparations. We compared only two groups of the data, namely, the control values ​​ with the values ​​in the presence of the test compound. asterisks indicatate values that differ significantly from the control values 

Comments on the Quality of English Language

English grammar and syntax should be carefully reviewed throughout the manuscript.

We found and corrected some errors in the text.

Sincerely,

Dr. N. Fedotcheva

Reviewer 2 Report

Comments and Suggestions for Authors

The authors investigated the influence of curcumin and cinnamic acid on isolated mitochondria. As shown in the work, curcumin and cinnamic acid sensitize mitochondria to open MPT pore, the first is predominantly due to partial depolarization, second is due to the activation of lipid peroxidation in mitochondrial membranes. The data help to explain the known contrary effects of curcumin on MPT pore opening as a mechanism of cell death. The research topic is relevant due to the application of both compounds in the synthesis of new drugs. An actual aspect is also their biotransformation, since cinnamic acid is both a microbial metabolite and a building block of the curcumin molecule.

Comments:

1.       It is necessary to mark significant differences in calcium retention capacity at different concentrations of curcumin in Figure 1F and to clarify description to Figure 3;

2.       In the Discussion section, a number of statements do not contain links to relevant sources; they must be indicated;

3.       Since the activity and biotransformation of these phenolic compounds is being compared, it is worth presenting their chemical structures, in the Introduction section.

Author Response

Thank you very much for your useful questions and comments. We have made corresponding corrections and additions to the manuscript.

Comments:

  1. It is necessary to mark significant differences in calcium retention capacity at different concentrations of curcumin in Figure 1F and to clarify description to Figure 3;

Replay:

We have added in Figure 1 asterisks (*) indicating values that differ significantly from the control values and expanded the legend to Figure 3.

  1. In the Discussion section, a number of statements do not contain links to relevant sources; they must be indicated;

Replay:

We have included the necessary links in the first paragraph of the Discussion.

  1. Since the activity and biotransformation of these phenolic compounds is being compared, it is worth presenting their chemical structures, in the Introduction section.

Replay:

We have added the chemical structures of curcumin and cinnamic acid to the table that compares their effects and where the structures are very relevant.

Sincerely,

Dr. N.Fedotcheva

Reviewer 3 Report

Comments and Suggestions for Authors

The manuscript presents a complex research on the mechanism of action for two natural phenolic compounds, curcumin and cinnamic acid, at the mitochondrial level.

The pharmacological effects of the curcumin, the main compound from turmeric, have been studied a lot and the results of these researches are multiple and varied. In this context, new studies to complete the data regarding the possible targets of action of these natural compounds are welcome.

The authors have carried out a very complex, very laborious, well documented and argued study, with complete information both in the documentation part and in the interpretation and discussion of the results.

A very important aspect in teh evaluation of pharmacological activity of curcuminis teh concentration tested, the authors very well underlining the possibility of changing the protective effect into a toxic effect, when the concentration of curcumin increases.

Author Response

Dear Reviewer, thank you very much for supporting our research.

Sincerely, Dr. N.Fedotcheva